# Treatment with Bacterial Biologics Promotes Healthy Aging and Traumatic Brain Injury Responses in Adult *Drosophila,* Modeling the Gut–Brain Axis and Inflammation Responses

**DOI:** 10.3390/cells10040900

**Published:** 2021-04-14

**Authors:** Brandon Molina, Jessica Mastroianni, Ema Suarez, Brijinder Soni, Erica Forsberg, Kim Finley

**Affiliations:** 1Department of Biology, Shiley BioScience Center, San Diego State University, San Diego, CA 92182, USA; bmolina@sdsu.edu (B.M.); jmfmastroianni@yahoo.com (J.M.); elsuarez@sdsu.edu (E.S.); 2Department Chemistry and Biohemistry, San Diego State University, San Diego, CA 92182, USA; brijssoni@gmail.com (B.S.); eforsberg@sdsu.edu (E.F.)

**Keywords:** gut–brain axis, inactive bacteriologic (IAB), probiotic, neural inflammaging, traumatic brain injury (TBI), Toll-like receptor (TLR), nucleotide-binding oligomerization domain-containing protein (NOD2), nuclear factor kappa-light-chain-enhancer of activated B cells (NF-κβ), pathogen associated molecular pattern (PAMP), anti-microbial peptide (AMP)

## Abstract

*Drosophila* are widely used to study neural development, immunity, and inflammatory pathways and processes associated with the gut–brain axis. Here, we examine the response of adult *Drosophila* given an inactive bacteriologic (IAB; proprietary lysate preparation of *Lactobacillus bulgaricus*, ReseT^®^) and a probiotic (*Lactobacillus rhamnosus*, LGG). In vitro, the IAB activates a subset of conserved Toll-like receptor (TLR) and nucleotide-binding, oligomerization domain-containing protein (NOD) receptors in human cells, and oral administration slowed the age-related decline of adult *Drosophila* locomotor behaviors. On average, IAB-treated flies lived significantly longer (+23%) and had lower neural aggregate profiles. Different IAB dosages also improved locomotor function and longevity profiles after traumatic brain injury (TBI) exposure. Mechanistically, short-term IAB and LGG treatment altered baseline nuclear factor kappa-light-chain-enhancer of activated B cells (NF-κβ) signaling profiles in neural and abdominal tissues. Overall, at select dosages, IAB and LGG exposure has a positive impact on *Drosophila* longevity, neural aging, and mild traumatic brain injury (TBI)-related responses, with IAB showing greater benefit. This includes severe TBI (sTBI) responses, where IAB treatment was protective and LGG increased acute mortality profiles. This work shows that *Drosophila* are an effective model for testing bacterial-based biologics, that IAB and probiotic treatments promote neuronal health and influence inflammatory pathways in neural and immune tissues. Therefore, targeted IAB treatments are a novel strategy to promote the appropriate function of the gut–brain axis.

## 1. Introduction

*Drosophila* is a classic genetic model that has been widely used to investigate genetic components that regulate neural development and aging, as well as the molecular mechanism involved with the highly conserved signaling pathways, which regulate proteolytic, innate immune, and inflammatory responses [1,2,3,4,5,6,7]. Our group has largely focused on the macroautophagy pathway’s role during neural aging and following exposure to stress, including fasting, oxidant exposure (H_2_O_2_), and traumatic brain injury (TBI) [4,7,8,9,10]. These in vivo studies have assessed the influence of genetic and environmental factors on healthy neuronal aging by examining the in vivo formation of neural protein aggregates and transcriptional changes, together with behavioral and longevity-based studies. Modified diets (intermittent fasting), drug treatments, and genetic factors can alter neural function and global health profiles [4,7,10,11,12,13,14]. More importantly, these studies have led to the development of a highly effective, integrated therapeutic testing platform that takes advantage of *Drosophila*’s well-characterized molecular–genetic tools, relatively short lifespans, and complex behaviors, which permit large numbers of flies to be examined across multiple ages [3,7,10,11,13,14].

Interestingly, both fly and human systems show similar patterns of metabolic and immune dysregulation, as well as responses to elevated oxidative stress (OS) and chronic inflammation, implying conserved mechanisms [1,3,7,9,10,15]. Understanding these conserved mechanisms in whole animals are critical when examining complex processes linked to aging, chronic disease states, and traumatic injury [1,3,16]. Recent studies have shown that even modest OS level changes, including the production of reactive oxygen/nitrogen species, can further exacerbate inflammation in the neural tissues [13,15,16,17]. Indeed, elevated OS levels, cytokine imbalances, and glial activation can further damage neural tissue, accelerating neurodegenerative disease [18,19,20,21,22]. Identifying treatments that lower OS levels and promote cellular homeostasis would be beneficial for a range of maladies related to ageing or progressive cellular damage (Appendix A) [13,16,17,23].

Recent studies have detailed the importance of intestinal microbiota on neuronal function [3,6,18,23,24,25,26]. Termed the gut–brain axis, these interactions have been examined in the context of mental health and neurodegenerative disorders, such as Alzheimer’s disease and TBI [16,18,25,26,27,28,29]. This work strongly indicates crosstalk between the central nervous system (CNS) and immune–inflammatory systems, as well as the intestinal tract (gut), which has fueled interest in bacterial-based treatments for neurological disorders and healthy aging (Appendix A) [6,17,18,26,29,30,31]. Recent Alzheimer’s disease and TBI-based studies indicate that probiotic treatments have positive outcomes for patients [18,27,30,31,32]. Complicating most human-based studies are compliance issues, leading to non-uniform administration and dosing regimens [28,30,33,34,35]. Therefore, how ingested probiotics illicit neurological responses requires a high throughput model organism, like *Drosophila,* that has established gut–brain axis interactions [3,6,22,25,26,36].

In this report, we examine a novel proprietary inactive bacteriologic (IAB; ReseT^®^), which retains defined pathogen associated molecular pattern (PAMP) activity, including conserved interactions with a subset of human Toll-like receptor (TLR) and nucleotide-binding, oligomerization domain-containing protein (NOD) receptors (Toll-2, Toll-4, and NOD-2) [5,36,37,38,39,40]. The IAB is not a classic probiotic, in that it is not alive. As a result, it does not have administration issues noted in prior studies, while it potentially maintains conserved interactions with the gut immune system. In this report, a series of in vivo Drosophila studies were conducted to examine the impact that IAB and probiotic treatments have on the mature gut–brain axis. Existing *Drosophila* neural aging and TBI platforms were used to look at health, immunity, and longevity profiles of adult flies [7,9,10,11,12,13,14,22,24,41,42,43]. This work served as a preliminary assessment of bacteriologic interactions with gut and neural tissues, providing unique insights in to conserved mechanisms that IAB or probiotic treatments have on gut–brain axis interactions, as well as the ability to lower inflammation and OS levels and promote healthy aging and neuronal function [16,18,28,32,37,44].

## 2. Material and Methods

### 2.1. Drosophila Stocks and Culturing Conditions

The Canton-S and *w^1118^* stock lines were originally obtained from the Bloomington *Drosophila* Stock center (Department of Biology, Indiana University, Bloomington IN 47405-7005, USA, https://flybase.org) and have been described previously [13,14]. Canton-S (CS) females were crossed with *w^1118^* males, and F1 offspring (*w^1118^*/+) used as wild-type (WT) controls for all studies [11,12]. Flies were collected 4 h following eclosion using CO_2_, aged in 25 flies per vial cohorts and maintained on standard fly media (molasses, corn meal, agar, baker’s yeast), using established culturing conditions that included 25 °C, 60–70% humidity, and a standard 12 h light/dark cycle [11,12].

### 2.2. IAB and LGG Media Preparations and Longevity Studies

The IAB treatment represents a proprietary, inert lysate preparation made from the inactivated *Lactobacillus bulgaricus* (Labyrinth Holdings, Houston, TX, USA). To establish a treatment protocol and define IAB dosage for adult flies, 0.5 g of the lysate powder was ground into a fine powder using an Omni Bead Ruptor-24 homogenizer (Omni International, Kennesaw, GA, USA) [9]. Deionized (DI) water (10 mL was added, and the resulting slurry was agitated overnight, stored at 4 °C, and used to make fresh working dilutions of the biologic. Depending on the assay, IAB dilutions were made in DI water, with final working dilutions ranging between 1:50 to 1:1600 from the primary slurry (50 mg/mL). Aliquots from each IAB dilution (100 µL) were pipetted into individual vials containing fly media (10 mL) and allowed to dry for ~2 h before adult flies (25–30 per vial) were added. For each pre-treatment or aging study fly, cohorts were turned on to freshly prepared media three times per week [11,12,13,14].

For comparative studies, live LGG (*Lactobacillus rhamnosus*) cultures were prepared from a single bacterial colony, cultured overnight (37 °C) in M-9 media (33.9 g/L Na_2_HPO_4_, 15 g/L KH_2_PO_4_, 5 g/L NH_4_Cl, and 2.5 g/L NaCl) [17,40]. The following day, the overnight culture was used to inoculate fresh media (5 mL), with bacteria being collected (2.0 OD), cells pelleted, washed (2×), and resuspended in fresh M-9 media at 10^5^ to 10^6^ colony forming units (CFUs) per mL. An aliquot of LGG bacteria (10^5^ CFUs) were placed in individual fly vials (8 mL media), evenly distributed, and allowed to air-dry for 2 h (RT). This LGG dosage corresponds to 10^10^ CFU per 80 kg (human) [45]. For comparative studies, IAB was diluted in M-9 media instead of DI water, and all fly cohorts were transferred to fresh media weekly (3×). For longevity studies, the number of dead flies for each condition was counted and recorded. This information was used to generate Kaplan–Meier survival plots, average lifespan (days) in statistical analysis, including *p*-values (Student’s *t*-test), using previously published techniques [9,11,13].

### 2.3. Negative Geotaxis Response

The *Drosophila* negative geotaxis response (NGR) involves mechanical stimulation of an innate escape response, using the previously described rapid iterative negative geotaxis protocol and apparatus (RING) [11,12]. Briefly, seven groups of 10–25 flies are tapped down, and digital movies/images taken after 5 s. Flies are allowed to rest for 1 min between three replicate runs. Digital images are analyzed, and the distance traveled (cm) for each fly and run are scored and assigned a value between 0 (bottom) to 6 (top). The triplicate run counts were used for this to establish the average climbing indexes for each assay cohort (65 to 125 flies), SEM, and statistical significance or *p*-values (*n*, student two-tailed *t*-test) for each cohort within a particular study [11,12].

### 2.4. Western Analysis

Fly cohorts (30 per condition) were treated with IAB (1:800) for one or three weeks before being collected, flash frozen, and stored (−80 °C). Tissue isolation, protein preparation, electrophoresis, and Western blot analysis followed established protocols. For aggregate analysis, isolated heads were sequentially extracted with established Triton-X and SDS buffers and techniques [8,10,12]. Blots were sequentially probed with anti-Tubulin (E7, Developmental Studies Hybridoma Bank, University of Iowa. Dept. of Biology, 028 BBE, 210 E. Iowa Ave, Iowa City, IA 52242, USA (DSHB), https://dshb.biology.uiowa.edu), anti-Ref(2)P and anti-ubiquitin (P4D1, Cell Signaling Technologies, Inc., 3 Trask Lane, Danvers, MA 01923, USA) antibodies [9,10,11,14]. For additional studies examining immune responses, see Appendix A. Blots were developed using Thermo Scientific West Dura Substrate (ThermoFisher Scientific, www.thermofisher.com, ChemiDoc digital Imaging System, and Quantity One software (Bio-Rad, 1000 Alfred Nobel Drive, Hercules, CA 94547 USA, v4.5) [10]. The intensities of individual protein bands were quantified using ImageJ (https://imagej.nih.gov, 2019) [8,9,10,12].

### 2.5. Traumatic Brain Injury

The TBI methods used for *Drosophila* have been previously detailed [10]. Briefly, flies from different genders, ages, or treatment regimens were anesthetized and placed in 2 mL screw cap tubes (10 flies/tube) [9]. Control and traumatized flies were allowed to recover from anesthesia before being placed in the Omni Bead Ruptor-24 homogenizer (Omni International, Kennesaw, GA, USA) and subjected to specific pre-programmed shaking/trauma conditions involving 5 s trauma bouts [9]. Following the completion of the injury, all fly cohorts are returned to vials containing control or treated media, allowed to recover, and maintained using standard care and culturing conditions [9]. For this study, fly cohorts were exposed to 10 mild bouts of trauma (mTBI; 10×, 2.1 m/s) or one severe trauma bout (mTBI; 1×, 4.3 m/s) [9].

### 2.6. Mortality Indexes

Acute mortality indexes were determined by subjecting young flies to a single 5 s bout of severe injury (sTBI; 1×, 4.35 m/s) [9]. Flies were placed on IAB-conditioned media, and the number of dead counted 24 and 48 h following trauma. Total fly counts from replicate vials (*n*) were used to calculate the average percent dead, SEM, and *p*-values [9].

### 2.7. Quantitative PCR

Young fly (one-week-old) cohorts, without treatment (controls) or following 1 or 2 days of IAB (1:800) or LGG (10^5^ CFUs) treatment, were collected, flash-frozen, and stored (−80 °C). Heads (30) or abdominal tissues (10) were isolated and RNA pools extracted (Trizol, ThermoFisher Scientific, Grand Island, NY, USA), and cDNA library synthesis (Thermo Scientific, Pittsburg, PA, USA) followed established protocols [7,9,10,11]. Quantitative PCR was performed on a CFX Connect Real-Time PCR Detection System (Bio-Rad) and Universal PCR SYBR Mix reagents (Bio-Rad) [9,10]. Specific primer sequences for the *AttC*, *DptB*, *Mtk*, and *Exba* (reference) genes are available upon request [9]. The Pfaffl method was used to quantitate Cq expression profiles, corrected for load (Exba) to establish corrected expression values. Relative mRNA levels of control fly cohorts were set at 1.0, and subsequent expression levels from different tissues or treatments used to calculate average normalized values, SEM, and *p*-values (two-tailed Student’s *t*-tests) [9,10].

### 2.8. Receptor Activation

The receptor activation profile of the IAB (Labyrinth Holdings, Houston, TX, USA) was confirmed by InvivoGen using their PRR Ligand Screening platform service (https://www.invivogen.com/multi-prr-ligands, San Diego, CA, USA) and blind-coded IAB samples [38]. This testing platform uses established engineered HEK293 cell lines that express individual human TLR or NOD receptors with known PAMP activity, which are linked to a classical reporter system (secreted embryonic alkaline phosphatase) [38]. The IAB was diluted in DI water (1:70) and tested together with matching positive and negative controls in triplicate. Averages compared to negative controls were used to assess the relative activation of each immune receptor type [38].

### 2.9. Statistical Analysis

Quantified behavior activity profiles, longevity, Western blots, and qRT-PCR graphs were generated using Microsoft Excel, and figures and tables were assembled in PowerPoint (Microsoft Office, Mac 2011 v14.7.1) [11,13]. Statistical analyses between groups were performed using the online GraphPad v.8 (https://www.graphpad.com) software and the Student’s *t*-test function (two-tailed, unpaired). All values are reported as mean values and SEM (+/−) [10,11].

## 3. Results

### 3.1. Negative Geotaxis Response

As an initial assessment of IAB’s impact on neural aging involved assays using the negative geotaxis responses (NGRs) or climbing behaviors of adult *Drosophila*. Analysis of climbing behaviors gives a rapid and consistent real-time assessment of acute neural function. Typically, there is a pronounced, age-dependent decline in NGR profiles, which can be suppressed in select genetic backgrounds or following interventions that promote longevity [10,11,12,13,14,24]. For this study, the baseline NGR profiles of one-week-old male flies were determined. Fly cohorts were then placed on media containing a range of IAB dilutions or dosages (0, 1:50, 1:100, 1:200, 1:400, 1:800 dilutions) for the remainder of the study [3,11,36]. Fly cohorts were placed on fresh IAB media three times per week, and NGR profiles determined weekly for weeks (five weeks of age). The results are illustrated in Figure 1A, with statistical details in Appendix A. Flies on standard food or on the lowest IAB dilutions showed normal, age-dependent decline in climbing behaviors. When compared to age-matched control, flies receiving 1:400 to 1:800 IAB treatments maintained their NGR profiles over the five-week study period. These findings were similar, but even more pronounced than those from intermittent fasting studies, where neural function and longevity were also promoted [10,11].

### 3.2. Longevity Profiles

Longevity studies were also conducted for flies receiving IAB treatments. Young flies were exposed to different IAB dosages starting at one week o, which continued throughout the entire study. The percent dead at each time point were used to produce the Kaplan–Meier Survival Profiles (Figure 1B) [11,12,13,14]. An additional lifespan study was conducted using higher IAB dilutions (1:1200 and 1:1600), with both studies summarized in Figure 1C and detailed in Appendix A. Consistent with the long-term maintenance of behaviors, the higher IAB dilutions significantly increased *Drosophila* longevity (Figure 1C and Appendix A). In contrast, flies placed on the lowest IAB dilution (1:50) showed a significant reduction in longevity, while 1:100 to 1:400 IAB-treated flies were unaffected (Appendix A). The IAB dilution with the greatest impact on longevity was the 1:1200 dilution (+23.1%), bracketed by improvements for the 1:800 (+10%) and 1:1600 (+15%) dilutions (Appendix A). When compared to other reported treatments, the maximal increase in mean lifespan profiles seen with 1:1200 IAB dosage is exceptional, and clarifies an effective therapeutic range for fly aging studies [12,13,14,20].

### 3.3. Neural Aggregate Changes Due to Aging

Other interventions that promote healthy *Drosophila* aging typically promote the long-term function of autophagy pathway in the CNS, as a consequence reducing or slowing the natural buildup of neural protein aggregates [9,14]. To examine the protein aggregate buildup, flies were treated with the IAB concentration with the greatest impact on longevity (1:1200 dilution). Treatment started at one or three weeks of age, and along with untreated controls, all flies were collected at four weeks of age and flash-frozen. Heads were sequential extracted, and insoluble protein fractions (SDS) Western blotted and probed for neural aggregate markers, including insoluble ubiquitinated proteins (IUPs) and Ref(2)P (human p62 homologue) (Figure 2) [7,8,11]. Corrected values were used to determine average protein aggregate levels. After only one week of IAB (1:1200) treatment, middle-aged flies showed the normal buildup of aggregates, while a three-week exposure did significantly lower neural aggregate levels (Figure 2) [7,8,11]. Together with preserving adult behaviors and promoting longevity, this indicates IAB treatment can promote normal proteolytic functions and healthy neural aging [11,12].

### 3.4. Neural Function Following Trauma

Traumatic brain injury and elevated neural inflammation has been closely linked to the development of progressive neurological disease [9,15,18,29,32,46]. To examine this risk factor, both a mild repetitive (mTBI) and severe (sTBI) traumatic brain injury *Drosophila* models were developed [9]. For this study, the ability of IAB treatments to preserve adult climbing behaviors (NGR) were assessed before and following mTBI-10× exposure. Initially, one-week-old flies were pre-treated for two or five days with defined IAB dosages (0, 1:400, 1:800, 1:1200) before mTBI-10× exposure (2.3 m/s), and returned to and maintained in matching IAB conditions (Figure 3A,B and Appendix A) [9]. In addition, flies were post-treated with defined IAB dilutions (Figure 3C and Appendix A). For each study, the NGR profiles were determined at one week (before) and two weeks of age following trauma exposure (Figure 3A–C and Appendix A). After mTBI-10× exposure, control flies showed a pronounced decline in climbing behaviors that was not related to aging [9]. Overall, most of the IAB treatments were highly protective and significantly helped to maintain climbing behaviors following trauma (Figure 3A–C, Appendix A) [9,11,12].

### 3.5. Aging Profiles Following Trauma

Previous studies have shown that mild repetitive trauma conditions (mTBI-10×) had a negative impact on adult *Drosophila* longevity profiles [9,42]. The lifespan profiles of one-week-old control and traumatized flies were examined, using the pre- and post-treatment protocols and select IAB dosages (0, 1:400, 1:800, 1:1,200). The Kaplan–Meier Survival profiles for controls and flies receiving two days of IAB pre-treated before mTBI-10× exposure are illustrated in Figure 4A,B. Longevity profiles for five-day pretreatment and post-treatment cohorts are shown in Figure 4C,D, and the statistical information for this study are included in Appendix A–C. Overall, flies pre-treated with IAB for two days (1:400) or five days (1:400, 1:800) showed a significant increase in average lifespans (Figure 4B,C, Appendix A), while post-treated fly cohorts had a slight decrease in longevity (Figure 4D, Appendix A) [12].

### 3.6. Conserved Aging and Trauma Responses

To determine whether other biologics could influence neural aging and trauma responses, we made direct comparisons between IAB and a commercially available probiotic, LGG [45,46]. For these studies, IAB and LGG preparations were suspended in M-9 broth, prior to application on fresh fly food. To determine that neither biologic altered feeding behaviors or nutritional intake (fasting), the weights of replicate fly cohorts were obtained before, 24 h, and 48 h after standard (control), M-9 media, LGG (10^5^ CFU, M-9), or IAB (1:800, 1:1200, M-9) treatments. A modest weight loss was detected for flies on control M-9 media. LGG and IAB treated flies did not show significant changes, indicating that flies found both treatments palatable (Appendix A) [3,12,47,48].

### 3.7. Basal Immune Profiles

The ability of both biologics to alter the innate immune system was initially examined for changes to downstream nuclear factor kappa-light-chain-enhancer of activated B cells (NF-κβ) targets in the fly abdominal tissues. Adult flies were placed on IAB (1:800, M-9) or LGG (10^5^ CFUs, M-9) media for 24 h, collected, and flash-frozen, and triplicate RNA pools were isolated from dissected abdomens (intestinal and immune tissues). Triplicate cDNA libraries were generated, and qRT-PCR analyses used to establish the expression levels of select *Drosophila* anti-microbial peptides (AMPs). Fly AMPs are directly regulated by NF-κβ signaling, and show dynamic expression changes with age, infection, or following trauma [9,49,50]. AMPs also have immuno-modulatory chemokine and cytokine-like functions, and reflect immune action and OS levels [9,36,51]. Both biologics significantly lowered *AttC* and *DptB* mRNA levels, while IAB treatment reduced the baseline *Mtk* mRNA levels (Figure 5A) [9]. Previous research showed that *AttC* and *DptB* expression is largely regulated by the *Drosophila* IMD (human TNF homologue) signaling pathway, while *Mtk* regulation is influenced by IMD and Toll receptor activity [2,41,51]. This indicates that both biologics likely alter IMD signaling (*AttC*, *DptB*), while IAB influences both systems in abdominal or gut tissues (*Mtk*, Figure 5A) [2,41,51]. The potential influence of IAB on neural NF-κβ signaling was also examined. Head RNA was isolated from control and IAB-treated fly cohorts (two days, 1:400) and used to generate triplicate cDNA libraries. The *Mtk, DptB*, and *AttC* expression profiles were assessed using qRT-PCR analysis (see Appendix A). While not of statistical significance, a similar downward expression trend was seen for each AMP. This indicates that IAB treatment impacts gut–brain axis function through NF-κβ pathway interactions, altering the regulation of conserved immune–inflammation factors [28].

### 3.8. Impact of Biologics on Behavior and Longevity

Given that each biologic treatment had unique impact on AMP expression profiles, we directly compared their impact on neural aging and longevity. The NGR profiles for young (one-week-old) or middle-aged (three-week-old) flies were obtained, followed by treating individual cohorts with control (M-9), LGG (10^5^ CFUs), or IAB (1:800, 1:1200) media for one week (Figure 5B, Appendix A). Untreated flies showed the normal NGR decline, while only one week of LGG or IAB exposure helped maintain climbing behaviors (Figure 5B, Appendix A). Longevity studies also revealed that early IAB treatment (one week, 1:800) promoted longevity, while middle-aged flies treated with either biologic had increased lifespans (Figure 5C, Appendix A) [3,21,22].

### 3.9. Mild and Severe Trauma Responses

Comparisons were also made for trauma responses. Fly NGRs were obtained following one week of IAB (1:800; 1:800 to 1:100) or LGG (10^5^ CFUs) treatment, and one and two weeks following mTB-10× exposure (Figure 6A, Appendix A). Exposure to both biologics helped maintain NGR profiles at each time point following trauma (Figure 6A, Appendix A). Previously, intestinal integrity assays (SMURF) have shown that mTBI-10× conditions do not alter the integrity of the *Drosophila* intestinal tract [9,52]. In contrast, exposure to severe trauma conditions (sTBI-1×, 4.3 m/s) lead to both intestinal leakage and an elevated 24 to 48 h mortality index (MI^24^, MI^48^, respectively) [9,52]. Initially, young flies pretreated with different IAB dosages (1:50, 1:100, 1:200, 1:400) and then exposed to sTBI-1× were examined. The number of dead flies were counted at 24 and 48 h after trauma and MIs were determined (Figure 6B). In this case, the effective IAB therapeutic dose was at 1:100 dilution, offering a significant level of protection (Figure 6B). Fly cohorts were also pretreated (two days) with IAB (1:100) or LGG (10^5^ CFU) before sTBI-1× exposure. Under these conditions, LGG flies showed a significant increase in MI^24^ and MI^48^ profiles when compared to control (M-9) or IAB (M-9)-treated cohorts (Figure 6C) [9,52].

### 3.10. In Vitro Receptor Interactions

To confirm the ability of the IAB to illicit pathogen-associated molecular pattern (PAMP) responses, the InvivoGen PRR ligand screening and immune activation platform was conducted, which uses a cell culture-based assay to examine TLR and NOD receptors agonist panels [38]. Triplicate IAB assays with matching controls were conducted, and cell lines expressing the human TLR2, TLR4, or NOD2 receptors showed a marked activation following IAB exposure (Table 1). Compared to negative controls, 4.0-fold or greater inductions are considered moderate agonist activity, while a 10-fold activation represents a highly significant PAMP-based response [38].

## 4. Discussion

Research has shown the importance of gut–brain axis activity for regulating complex physiological processes, ranging from aging to inflammation and the management of oxidative stress [16,29,30,38]. Studies have largely focused on comparing health associations with the composition of the resident gut metagenome, using high-throughput sequencing and bioinformatic techniques [29,30,48,53,54]. However, the nature of these studies fails to address the complex, whole animal, real-time interactions that bacteriologics may have with the gut–brain axis [3,29,30,48,55]. The exact nature of the communication between gut biologics, dynamic OS fluctuations, and host neural function is still being defined. Therefore, characterizing conserved interactions between biologics with the innate immune system of a whole animal model system provides unique opportunities to examine the gut–brain axis’ impact on complex physiological responses.

Historically, *Drosophila* studies have been instrumental for the identification, characterization, and molecular dissection of conserved Toll, NF-κβ, and NRF2 pathway components [22,35,36,37,39,40]. The functional similarities between the fly and human innate immune systems are now well-established at the level of molecular, cellular, and tissue responses, which are depicted in Appendix A [2,3,26,30,31]. This makes *Drosophila* an ideal model to quickly compare different biologic treatments and their impact on physiology and integrated gut–brain axis processes. We anticipated the fly intestinal tissues would interact with bacteriologics, such as an IAB or probiotics, via the conserved Toll and IMD signaling pathways [5,30,42,43]. An additional advantage of *Drosophila* aging and trauma studies is the ability to quickly test a range of treatments with large numbers of flies, and detect subtle but significant changes to behavior and longevity profiles, as well as changes to neural aggregates and immune expression profiles. This further clarifies the impact of the gut–brain axis on target tissues [8,9,11,12].

This report focused on administering a novel IAB with a defined PAMP repertoire (Table 1) and LGG to *Drosophila*, in order to further understand the role of a conserved immune communication mechanism in gut–brain axis responses. Findings from neural aging and TBI-based testing platforms show the marked benefits that oral consumption of IAB or LGG biologics have on longevity, neural health, and trauma responses in adult flies. These results provide an initial assessment of biologics and their impact on the communication of the gut–brain axis with neural systems. Aging studies first identified positive behavioral responses and longevity outcomes over a range of IAB dosages (Figure 1A,B). Together with LGG studies, this indicates that the type of biologic being delivered may have different gut–brain axis communication (Figure 1, Figure 3, Figure 4, Figure 5 and Figure 6). Since LGG is alive, it cannot use the standard PRR screening assay. However, the InvivoGen platform uses whole-cell, heat-killed *Lactobacillus rhamnosus* bacteria as a TLR2 agonist (https://www.invivogen.com/hklr, accessed on 20 March 2015), suggesting that live bacteria would have the same functional activity [39,40]. Therefore, the unique agonist profiles (PAMP) for each treatment likely correspond to the subtle fly longevity, behavior, and targeted trauma response differences detected in these studies. Selection of PAMP profiles could provide an unexplored treatment platform designed for healthy aging and mitigate neural aging and immune dysfunction [39,41,51].

Modulating the age-related decline in neural autophagy and protein aggregate profiles (Figure 2) is consistent with the ability of IAB to promote more behaviorally youthful profiles (Figure 1A). Baseline expression changes to NF-Κβ gene targets (AMPs) in key tissues (gut, brain) following the administration of biologics were also identified (Figure 5A, Appendix A). This further confirms that conserved gut–brain axis interactions occur in IAB- and LGG-treated *Drosophila*, changing NF-κβ signaling and helping to promote healthy aging and TBI responses (Figure 5A, Appendix A) [9,24,42,43]. To more directly assess NF-κβ responses, Western analysis was used to examine the NF-κβ pathway inhibitor, I-κβ (Cactus) in fly neural tissues (Appendix A) [2,44]. Pathway activation requires the rapid proteosome turnover of I-Κβ potentially serving as a dynamic marker for NF-Κβ and proteolytic activity [2,39]. When compared to untreated controls, aged IAB treated flies showed more youthful I-κβ fluctuations following acute pathway activation (mTBI-10×; Appendix A). While the I-κβ findings are preliminary, taken together with age-dependent changes to neural autophagy profiles and baseline AMP expression levels, they indicate that select biologics can modulate conserved NF-κβ pathway interactions and impact gut–brain axis functions [9,42,43].

Additional insights into the benefits of bacteriologic treatment regimens were highlighted by the positive impact that select IAB and LGG dosages had on traumatized flies. Climbing behaviors were preserved in flies exposed to mTBI-10× using pre- or post-treatment protocols (Figure 3A–C and Figure 5B; Appendix A). IAB also had a positive impact on longevity impaired by trauma (Figure 4A–D, Appendix A). This suggests that along with basal- and age-related changes to the gut–brain axis, trauma adds an additional level of complexity to inflammation, which can be modified through oral PAMP administration (Figure 6A, Appendix A). Previous mammalian probiotic studies have alluded to these benefits [18,26,28,29,37,44,55]. Severe trauma studies have also detected a marked difference between flies receiving IAB or LGG treatments (sTBI-1×, Figure 6B,C). Under these conditions, each biologic had opposing effects on trauma-related death, with LGG significantly increasing acute mortality rates (MI^24^ and MI^48^; Figure 6C) [9]. This suggests that the positive impact that bacteria may have on immune signaling and gut–brain axis function could be overwhelmed by secondary infections and sepsis-related complications [9,15,41].

In conclusion, the *Drosophila* aging and trauma-based studies presented here validate that high-throughput *Drosophila* models can differentiate the effects of different bacteriologic treatments (IAB and LGG) on the gut–brain axis and provide quantifiable information on linked physiological responses. The gut–brain axis responses of longevity, neural health, and trauma mitigation were initiated by conserved intestinal TLR and NOD immune signaling cascades via the activation by oral PAMP administration [39,41,51]. Although the activation of TLR and NOD receptors by PAMPs is not new, the oral administration of select PAMPs from IAB to promote beneficial gut–brain axis responses is novel, and opens the door to a gut–brain axis treatment platform targeting healthy aging and mitigation of acute trauma impacts [9,22]. Additional beneficial phenotypes could be identified through expanded studies on different PAMP profiles, and dosage regimes would be beneficial. Finally, the preliminary results on IAB’s impact on the age-related dysregulation of the fly immune system requires further investigation. Implications of a therapeutic delay or potential reversing of immune system dysfunction has vast implications for the treatment of chronic human diseases, including neurodegeneration [22,36,37,39,40,47].

## Figures and Tables

**Figure 1 cells-10-00900-f001:**
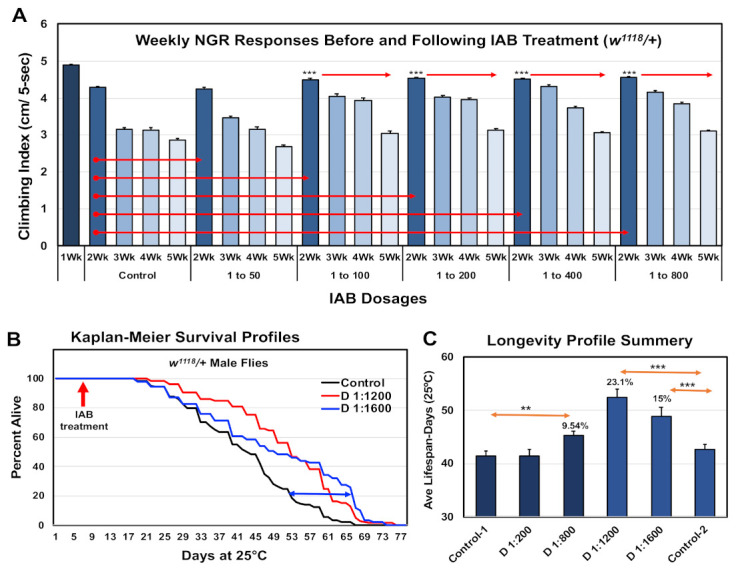
Therapeutic impact of inactive bacteriologic (IAB) dosages on *Drosophila* neural function and longevity. (**A**) Negative geotaxis responses of controls (0) and male fly cohorts exposed to IAB at 1:50, 1:100, 1:200, 1:400, or 1:800 dilutions. Starting at one week of age, climbing indexes were determined weekly (Wk). (**B**) Longevity profiles of fly cohorts exposed to media containing 0, 1:1200, or 1:1600 IAB dilutions. (**C**) Average and SEM summary profiles of two independent longevity studies (** *p*-values < 0.01, *** *p*-values < 0.001). See Appendix A for fly numbers (*n*) and additional information.

**Figure 2 cells-10-00900-f002:**
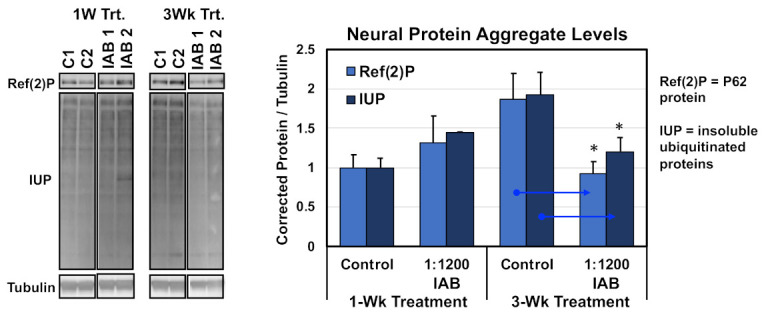
Western analysis of protein aggregates following IAB treatments. Insoluble ubiquitinated proteins (IUP) and Ref(2)P aggregate marker profiles in aged neural tissues (four weeks) after one or three weeks of IAB treatment (1:800 dilution). * *p* < 0.05.

**Figure 3 cells-10-00900-f003:**
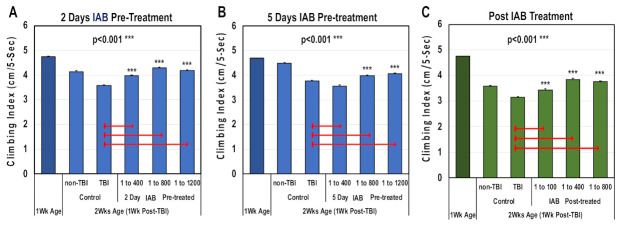
The negative geotaxis responses (NGRs) of young *Drosophila* before and after trauma. Geotaxis responses of flies before and after (**A**) two days or (**B**) five days of IAB pretreatment (0.0, 1:400, 1:800, 1:1200) and one week following mild traumatic brain injury or mTBI-10×. (**C**) The NGR profiles of flies exposed to mTBI-10× and post-treated with IAB (0.0, 1:100, 1:400, 1:800). (*p* values *** < 0.001). See Appendix A for fly numbers (*n*) and additional information.

**Figure 4 cells-10-00900-f004:**
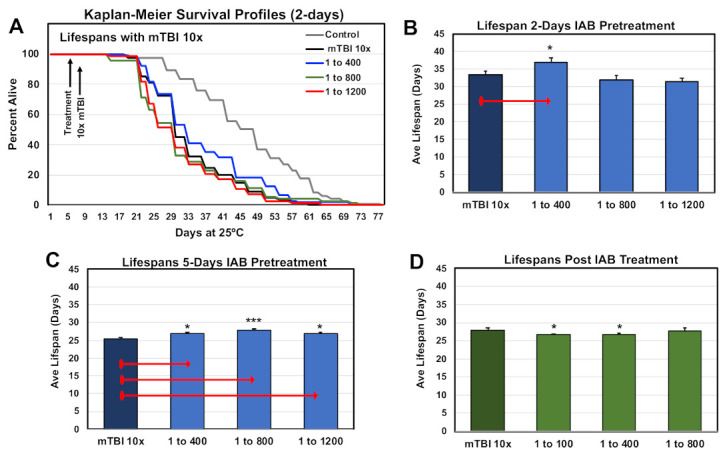
Impact of biological treatment on the longevity profiles of traumatized *Drosophila*. (**A**) The Kaplan–Meier Survival profiles of male flies pre-treated (two days) with different IAB dosages. (**B**–**D**) Average lifespan profiles of fly cohorts pre-treated for (**B**) two days or (**C**) five days before mTBI-10× exposure, and (**D**) post-treated profiles. (* *p*-values <0.5, *** *p*-values < 0.001). See Appendix A–C for fly numbers (n) and additional information.

**Figure 5 cells-10-00900-f005:**
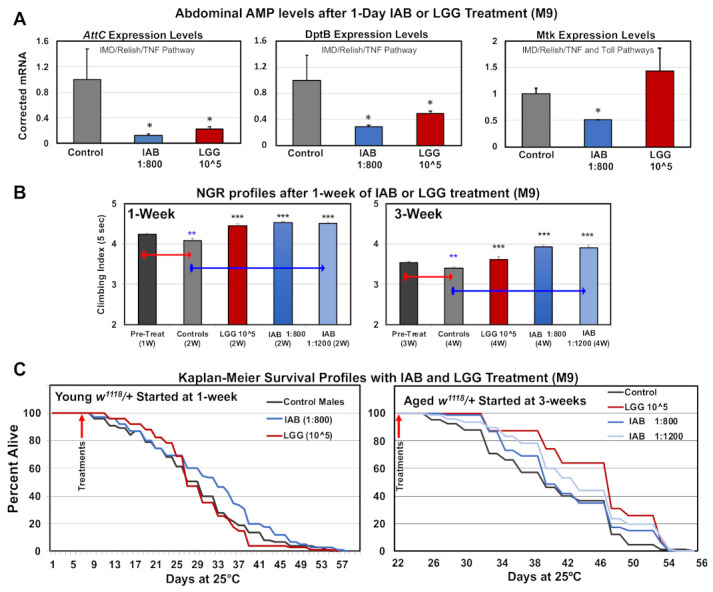
Immune and age-related responses to IAB and *Lactobacillus rhamnosus* (LGG) treatments. (**A**) Anti-microbial peptide (AMP) expression profiles in fly abdominal tissues following 24 h of IAB (1:800) or LGG (10^5^ CFU) treatment. (**B**) NGR of young (one-week-old) or middle-aged (three-week-old) flies after one week of control (M-9), LGG (10^5^ CFUs), IAB 1:800, or IAB 1:1200 treatment. (**C**) The Kaplan–Meier Survival profiles of young or middle-aged flies with control (M-9), IAB 1:800, IAB 1:1200, or LGG (10^5^ CFUs) treatment (* *p*-values < 0.5, ** *p*-values < 0.01, *** *p*-values < 0.001). See Appendix A for fly numbers (n) and additional information.

**Figure 6 cells-10-00900-f006:**
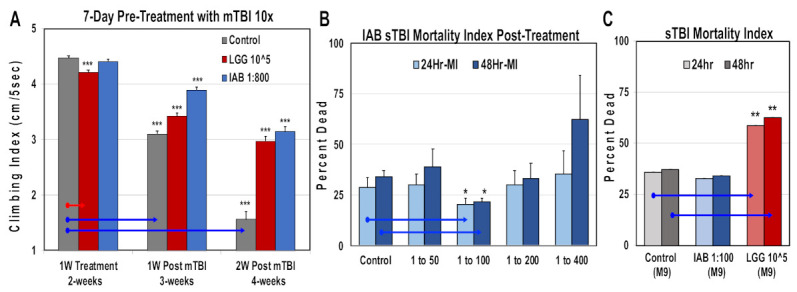
Impact of biologic treatments on trauma responses. (**A**) Geotaxis profiles of flies pre-treated with LGG (10^5^ CFUs) or IAB (1:800), as well as one and two weeks following mTBI. See Appendix A for additional information. (**B**) Young flies were given different IAB dilutions two days before sTBI-1× exposure. Dead flies were counted after 24 h and 48 h, and used to calculate mortality indexes (MIs). (**C**) The MI profiles of control (*n* = 78) flies, as well the pre-treated (two days) with IAB (1:800, *n* = 79) or LGG (10^5^ CFUs, *n* = 75) fly cohorts. (* *p*-values < 0.5, ** *p*-values < 0.01, *** *p*-values < 0.001). See Appendix A for additional fly numbers (*n*) and information.

**Table 1 cells-10-00900-t001:** InvivoGen TLR (Toll-like receptor) and NOD (nucleotide-binding, oligomerization domain-containing protein) receptor activation assay.

TLR/NOD	Negative Control		IAB Powder		Fold
Receptor	Average ^#^	Std Dev	Average ^#^	Std Dev	Induction *
hTLR2	0.052	0.001	2.521	0.025	48.5
hTLR3	0.127	0.002	0.535	0.014	4.2
hTLR4	0.163	0.027	1.877	0.076	11.5
hTLR5	0.066	0.001	0.173	0.003	2.6
hTLR7	0.095	0.002	0.119	0.014	1.3
hTLR8	0.110	0.003	0.125	0.001	1.1
hTLR9	0.156	0.011	0.204	0.003	1.3
NOD1	0.070	0.002	0.107	0.000	1.5
NOD2	0.073	0.001	1.649	0.046	22.6

# Averages represent results from three independent assays. * Fold induction is the IAB response values divided by the negative control values.

## Data Availability

Data sets outlined in the report and Appendix A section are original data sets. PPR studies were conducted by InvivoGen and raw data provided by Labyrinth Holdings, Houston, TX, United States.

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
