# Peer review of "Treatment with Bacterial Biologics Promotes Healthy Aging and Traumatic Brain Injury Responses in Adult Drosophila, Modeling the Gut–Brain Axis and Inflammation Responses"

_cells, 2021, doi:10.3390/cells10040900_

Round 1

Reviewer 1 Report

This manuscript by Molina et al. investigates the role of probiotics and an inactive bacterial product on neuronal activities of Drosophila. The authors address two models of neurological stress including aging and TBI and attempt to draw specific mechanistic connections driving the gut-brain-axis. Overall, the data presented is interesting and clinically relevant. However, the authors need to elaborate more on how this manuscript strictly addresses a gut-brain-axis mechanism or reword the manuscript to “suggest” such a connection. Including extra information on the downstream responders of the direct receptor signaling presented at the end of the manuscript would be a great strength to these mechanistic principles. Finally, the connections to the immunological regulation of the IAB and LGG is weak and should be expanded to all of the models studied.

Title:

  • Should be reworded. As written, suggests that Bacterial biologics promote TBI. Also Gut:Brain Axis should be written as Gut-Brain-Axis, without the colon and normalized as such throughout the text.

Abstract:

  • The nature of IAB should be described briefly, as this is not a common reagent.
  • All acronyms should spelled out at the their first mention.
  • The in vitro system should be expanded upon. Where these human cells?

Introduction:

  • Line 50: This is valid background information, however, explicitly indicating the models used to draw these conclusions would be useful. Was this is humans? Mice?
  • Inclusion of some information of the neurological comparison between Drosophila and human/mouse model would be beneficial. Since Drosophila do not have the same neural-immune architecture as mice/humans, it would be good to give some background on the similarities and differences.
  • Also, inclusion of more of the neurological symptoms of TBI in Drosophila compared to humans would be useful. And, if available, a reference indicating that Drosophila can be a good model of TBI and the potential limitations of the model.
  • Line 74/75: This needs to be reworded. Specifically indicating the Drosophila equivalents of the TLR and NOD receptors referred to.
  • Expanding, if possible, the nature of the IAB PAMP or its downstream effects in the in vitro studies would be beneficial.

Methods:

  • The contents of the Drosophila media used should be included as sugar content dramatically impacts NFkB signaling. Also, indicate whether active yeast cultures were used, as this can impact the gut microbiome of the the flies.
  • Elaborate with exposure to “10x mTBI” refers to.
  • Line 168: include the reference
  • The “Receptor Activation” section should be greatly expanded as the details of the this approach are very important.
  • In general, the authors used whole-head homogenates, and not just brain homogenates. This could skew data or also deemphasize significant contributions.

Results:

  • In general, the number of flies and tests used for each experiment should be included in the legend. Since the error bars seem very tight, this information is critical for assessing the data.
  • In the NGR, this is not only an metric of neurological activity, but also motility, which naturally declines with age. So, this metric could be a general output of “fitness” which should be outlined in this section.
  • Fig 1a: Authors describe a beneficial impact at 1:400 IAB dilution, while the 1:100 also seemed effective. In addition, is it normal that at the lower concentrations of IAB, there is a great impact? Where the higher concentrations having a toxic effect?
  • Why were different dilution subsets used for the longevity and NGR tests?
  • 2a: Indicate which bands are significant for this analysis and subsequently quantified.
  • Statistics on the Kaplan-Meier curves should be included comparing treatment groups to the control groups.
  • 4: The changes as observed in the plots seem extremely minor. How are these statistics being calculated, i.e. how are the flies/groups being pooled to give power to the analysis?
  • The data in Fig 2b are not described in order of the text. Figures or text should be arranged to be presented in a logical order.
  • Also, NFkB and the other outputs in Fig. 2b are not only immune correlates, but also reflect the general metabolism of the flies. The authors should rephrases this analyses’ conclusions with this frame of mind.
  • In addition, the data from the western blots does not seem to reflect the data. This follows that the fold change of data presented is very small, questioning the biological relevance of the results.
  • Section 3.7: Why would the flies on control media lose weight? Also, a larger timeline, up to 1 week would be more reflective of their general weight gain/loss in response to the treatment.
  • 5c: The right panel suggests that at 3 weeks, when the treatment was administered, no flies had died. It may be more representative to show the survival from day 0, then shown the differences after the late onset of treatment.
  • In section 3.8, why was the dose of 1:800 selected, when 1:400 IAB showed beneficial effects in the TBI experiments.
  • 5: The decrease in immune markers after 1 day in unchallenged flies is surprising. It could be a stress response rather than bona-fide changes in immune signaling since only 1 day post media change was investigated. This figure may be more appropriate as a supplemental to indicate that an immune response was not induced due to the probiotic or IAB treatment.
  • Section 3.11: This is a good assay, but comparative analysis of the equivalent receptors in the flies would strongly support the mechanistic insights. Qpcr of downstream targets from the samples already collected would be a great addition.

General Comments:

  • IAB seems to be slightly toxic to flies. The higher doses imposed less impact on longevity and the longer pretreatment did not benefit the TBI responses. This potential caveat should be addressed in the discussion. Following up on this, were the flies mass for all concentrations of IAB and LGG recorded? This would indicate whether IAB impacts the general fitness of the flies. This data should be included as a early supplemental figure, possibly supporting the dose data.
  • The caveat of the intestinal damage in response to the TBI model should be discreetly addressed. It is possible that the IAB and LGG are providing neural-like protection by reinforcing the intestinal walls.
  • Data would also be made stronger if the immunological markers were monitored in both model, aging and TBI.
  • The immune response was focused primarily in the abdominal sections. Biochemical markers in the brain of the flies would also be useful to establish the stronger gut-brain-axis connection.
  • Grammar and spelling needs to be carefully addressed throughout the manuscript. Perhaps editing by a native English speaker or editing service should be implemented.

Author Response

The reviewers had similar critiques. Therefore, our responses were combined.

Title, Abstract, Key Words and Introduction:

  1. The Gut:Brain:Axis has been changed to Gut-Brain-Axis throughout the paper.
  2. Title now reads: Treatment with Bacterial Biologics Promotes Healthy Aging and Traumatic Brain Injury Responses in Adult Drosophila, Modeling the Gut-Brain-Axis and Inflammation Responses.
  3. Abstract now reads inactive bacteriologic (IAB, proprietary lysate preparation of Lactobacillus bulgaricus, ReseT®).
  4. Key Word and Acronyms have been revised:
  5. Abstract has been revised and now reads: In vitro, the IAB activates a subset of conserved TLR and NOD receptors in human cells.
  6. Line 50 has been revised: From Drosophila models to human systems, metabolic or immune dysregulation and excessive oxidative stress (OS) as well as chronic inflammation have all been implicated as causal agents for the aging process, multiple chronic disease states and negative outcomes due to traumatic injury.
  7. Both Reviewers request additional clarity on using Drosophila as a model to examine bacteriologics efficacy. Therefore, Paragraph 4 and text throughout the manuscript text has been significantly revised. A new Figure S1 showing mito-ROS levels in fly GI and CNS tissues was included in Supplemental Information.
  8. Line 74/75 in Paragraph 4 has been revised.

Methods:

  1. The Drosophila media is fully cited in this report and Finley publications that examined the impact genetics, modified diets, drug treatments or trauma has on fly aging and stress profiles. The text has been revised.
  2. The mTBI-10x and sTBI-1x are further detailed and now written: For these studies, fly cohorts were exposed to ten mild trauma bouts (mTBI-10x at 2.3 m/sec) or one severe trauma bout (sTBI-1x at 4.1 m/sec). The use of Drosophila TBI models are extensively referenced throughout the text.
  3. The Reference in Line 168 has been included.
  4. Table 1. The “Receptor Activation '' studies were expanded in Methods and throughout the text. It was also noted in the Discussion are was noted that the InvivoGen platform uses whole cell heat killed Lactobacillus rhamnosus bacteria as a TLR2 agonist control (https://www.invivogen.com/hklr), suggesting that live bacteria would have the same functional activity.
  5. The method used to prepare whole fly head homogenates for Western or qRT-PCR analyses is noted in the text and extensively referenced.  Fly heads are estimated to contain ~95% neural and glial cell types, with ~5% from cuticle (inert), tracheal (inert), muscle, endocrine-adipose tissues.  Rapid flash freezing of flies and isolation of whole heads permits preparation of pristine RNA or protein samples essential to detect changing expression (qRT-PCR, RNA-seq), protein or aggregate (neural) profiles. 
  6. Flies were anesthetized using a CO2 diffuser.  This has been noted in the text.

Results:

  1. To clarify fly numbers used for each assay, additional data Tables have been included in Supplemental Data and noted in the text. Longevity and geotaxis (triplicate runs) assays include 65-150 individuals per cohort for each condition or time point. Similar statistical significance has been reported for fly aging, treatment or trauma-based studies.  A major advantage of Drosophila research is working with large cohorts of individuals.
  2. The NGR is largely a neurological metric and is widely used as a marker of fitness and aging in flies. The behavior declines starting at 2-weeks of age, coinciding with the buildup of neural aggregates. Fly muscles decline at ~4 to 5-weeks of age. To bacteriologic treatments this report focuses on young to middle-aged flies.
  3. How the IAB was diluted was revised throughout the text. When starting a novel treatment, a series of concentrations are examined to establish a therapeutic range. For this report, flies were treated with an equal volume of diluted IAB placed on media (Figure 1A). Physical parameters being assessed can also influence the optimal therapeutic dosage, treatment length and other factors.
  4. Figure 1A-B statistics are now included in Supplemental Table S1A-B.
  5. Figure 2B. We agree, the Westerns showing acute NF-Kb responses were a preliminary finding. This information was moved and reorganized as Supplemental Figure S4, to show I-Kb (Cactus) changes. These findings are now included in the Discussion. The Results Section numbering has been changed.
  6. Figure 4 the statistical breakdown is now included in Supplemental Tables 3A-B and noted in the text.
  7. Section 3.7. Figure S2. Studies indicated that adult Drosophila find ammonia to be noxious (NH4Cl), raising concerns about comparing IAB or LGG treatments diluted in M9 media. Acute weight changes were included since repeated anesthesia (CO2) can substantially impair adult flies and feeding behaviors.
  8. Figure 5C, examining older flies is a standard technique, used to examine treatments or damage across a range of ages. Kaplan-Meier curves often show an abbreviated timeline, since dead flies are not included.
  9. Section 3.8 the IAB 1:800 dosage was selected due to previous aging, NRG and mTBI findings.
  10. Figure 5A: To address both Reviewers comments on the abdominal AMP profiles (immune-inflammatory markers), we included Supplemental Figure S3, comparing head expression profiles of IAB treated flies. The literature also confirms the timeline that microbes have on AMP signaling in the fly gut.
  11. Section 3.11: Reviewers comment that the InvivoGen assay was informative was appreciated. Comparisons between fly-human receptor equivalents were expanded upon throughout the text.

Reviewer 2 Report

The manuscript by Molina and colleagues studies the role of bacterial biologics supplementation on immune response, age-related locomotor deficits, and longevity in Drosophila. The potential effects of bacteriologic and probiotic are of great interest.

  1. Page 3, line 106: “allowed to dry for ~2-hr before adult flies”. Mention the age of adult flies.
  2. The “GUT:Brain axis” should be changed as “gut-brain axis” throughout the paper.
  3. The title of the paper should be modified to better reflect the study findings.
  4. The abstract needs to be modified to provide an accurate summary of the research objectives, key methods, principal findings, and study conclusions.
  5. Page 3, line 101: “DI water”. What does this mean?
  6. Page 4, line 150: What does the experimental group Control group represent? Do the flies in the Control group placed in the Omni Bead homogenizer but not subjected to shaking? The authors should clearly explain the experimental procedures.
  7. Page 4, line 150: “flies were allowed to recover from anesthesia”. Mention the anesthesia used to anesthetize the flies.
  8. Do authors have suitable justification for not performing inflammatory cytokine assays to support the inflammatory hypothesis studies in the present report?
  9. The reviewer appreciates the work on the fly TBI model. The authors are suggested to discuss the secondary injury cascades are thought to account for the development of many of the neurological deficits observed after TBI. The paper (Int J Mol Sci 2020;21(2):588) should be discussed.

Author Response

The reviewers' reviews had similar comments. They are addressed in detail in the in the attached document.

Round 2

Reviewer 2 Report

The authors have satisfactorily addressed all the concerns raised by this reviewer.